# Changes in pneumococcal vaccine coverage in the Canadian Longitudinal Study on Aging (CLSA): An analysis based on the 2018–2021 follow-up 2 survey

Giorgia Sulis[1,2☯*], Nawal Maredia[1☯], Christina Wolfson[3,4], Nicole E. Basta[3]

**1** Faculty of Medicine, School of Epidemiology and Public Health, University of Ottawa, Ottawa, Ontario, Canada, **2** Methodological and Implementation Research Program, Ottawa Hospital Research Institute, Ottawa, Ontario, Canada, **3** Department of Epidemiology, Biostatistics and Occupational Health, School of Population and Global Health, Faculty of Medicine and Health Sciences, McGill University, Montreal, Quebec, Canada, **4** Neuroepidemiology Research Unit, Research Institute of the McGill University Health Centre, Montreal, Quebec, Canada

☯ These authors contributed equally to this work.
\* gsulis@uottawa.ca

## Abstract

### Introduction

Pneumococcal vaccination is recommended for older adults and individuals with chronic medical conditions (CMC) due to the high risk of invasive pneumococcal disease in these groups. Despite this, vaccination coverage in Canada remains below the national target of 80%, to be achieved by 2025. We conducted a new analysis of recently released data from the Canadian Longitudinal Study on Aging (CLSA), aimed at providing estimates of pneumococcal vaccine coverage from 2018–2021 among eligible adults, identifying sociodemographic disparities, and exploring changes over time since 2015.

### Methods

The CLSA, a nationally representative cohort launched in 2011, recently released data collected during the second follow-up visit (FUP2; 2018–2021). We conducted a cross-sectional analysis of participant self-reported pneumococcal vaccination status, stratified by sociodemographic characteristics, receipt of influenza vaccine in the previous 12 months, and contact with family doctor in the previous 12 months. Logistic regression was used to identify factors associated with being newly vaccinated for pneumococcal disease as reported during FUP2 compared with three years earlier during follow-up 1 (FUP1; 2015–2018). We previously reported pneumococcal vaccination estimates for 2015–2018.

---

**Data availability statement:** Data are available from the Canadian Longitudinal Study on Aging (www.clsa-elcv.ca) for researchers who meet the criteria for access to de-identified CLSA data. For all approved users an interinstitutional CLSA Access Agreement is signed that prohibits the sharing of the CLSA data beyond the approved research team. Information about how to access the data via the CLSA data application process can be found at https://www.clsa-elcv.ca/data-access/.

**Funding:** Funding for the Canadian Longitudinal Study on Aging (CLSA) is provided by the Government of Canada through the Canadian Institutes of Health Research (CIHR) under grant reference: LSA 94473 and the Canada Foundation for Innovation, as well as the following provinces, Newfoundland, Nova Scotia, Quebec, Ontario, Manitoba, Alberta, and British Columbia. The author(s) received no specific funding for this work. However, GS holds a Tier 2 Canada Research Chair in Communicable Disease Epidemiology and NEB holds a Tier 2 Canada Research Chair in Infectious Disease Prevention. The funders had no role in study design, data collection and analysis, decision to publish, or preparation of the manuscript.

**Competing interests:** The authors have declared that no competing interests exist.

## Results

Only 56.8% (95% CI: 55.8–57.7%; n = 10,530) of eligible study participants aged 65 years and older and 19.3% (95% CI: 18.1–20.5%; n = 4,055) of those aged <65 years with at least 1 chronic medical condition reported having received the pneumococcal vaccine when surveyed between 2018–2021. Males, rural residents, and individuals in certain provinces reported lower vaccination rates. Compared to three years prior, 28.4% of participants aged 65 years and older and 11% of participants aged <65 years with at least one CMC reported being newly vaccinated. Higher odds of being newly vaccinated were observed among individuals who reported having received influenza vaccination in the previous 12 months in both age groups.

## Conclusions

Pneumococcal vaccine coverage among Canadian adults aged 65 and older enrolled in the CLSA increased by only 2% between 2015–2018 and 2018–2021, and no changes were observed among those under the age of 65 with underlying conditions.

---

## Introduction

Pneumococcal disease, caused by *Streptococcus pneumoniae*, is a significant contributor to global morbidity and mortality, particularly among young children, older adults, and people with compromised immune systems or chronic conditions [1,2]. The bacterium can cause non-invasive diseases such as non-bacteremic pneumonia, otitis media, and rhinosinusitis, and can also lead to invasive pneumococcal disease (IPD) when it invades normally sterile sites like blood, cerebrospinal fluid, pleural fluid, joint fluid, or pericardial fluid [3,4]. IPD manifests as bacteremic pneumonia, sepsis, or meningitis, and has a high mortality rate; in adults, estimated mortality from invasive pneumococcal pneumonia has remained around 20% over the past 60 years [5]. In Canada, approximately 3,000 cases of IPD are reported annually, primarily affecting children under 5 and adults over 65 years of age [6]. The incidence of IPD in 2021 was 5.6 cases per 100,000 population, notably lower than the pre-COVID-19 pandemic incidence, which ranged from 9.0 to 10.9 cases per 100,000 population between 2009 and 2018 [7]. The 2021 incidence rate was similar to 2020 (5.9 cases per 100,000 population) but rose to 10.23 cases per 100,000 people in 2022 [7,8]. Among older adults, the burden is particularly elevated, with an annual incidence of 23.6 cases per 100,000 during the period of 2011–2015 and 24 cases per 100,000 in 2017 for those aged 65 years or older [9]. Provincial data suggest an even higher annual incidence among those over 85, with 57.5 cases per 100,000 compared to an overall average of 10.8 per 100,000 for all age groups in Ontario between 2010 and 2018 [10,11].

In response to the growing burden of pneumococcal disease, several countries across the globe have integrated adult pneumococcal vaccination into their National Immunization Programs as part of broader preventative health strategies

[12]. In 2024 Health Canada also recommended that adults aged 65 and older, and those aged 50–64 years at higher risk, receive a single dose of the pneumococcal conjugate 20-valent (Pneu-C-20) or 21-valent (Pneu-C-21) vaccine, irrespective of their previous pneumococcal vaccination history [13]. Immunocompromised adults aged 18–49 years are also advised to receive a dose of Pneu-C-20 or Pneu-C-21 based on individual risk assessments [13]. These updated recommendations aim to enhance protection against the evolving array of pneumococcal serotypes. It should be noted, however, that pneumococcal vaccination has been recommended for high-risk adults in Canada since 1989, when the 23-valent pneumococcal polysaccharide vaccine (PPV23) started being deployed through publicly funded programs [14].

Globally, pneumococcal vaccine uptake among older adults varies widely but is generally low, with coverage ranging from approximately 18% in several European countries to around 59% in the United States and 37.8% in Japan, despite national immunization programs and public funding support [12,15]. In countries with publicly funded healthcare systems, such as the United Kingdom, coverage reaches 71.5% [12]. According to the Seasonal Influenza Vaccination Coverage Survey for 2020–2021 carried out by the Public Health Agency of Canada, out of 2739 adults aged 18 years and above surveyed only 839 (55%; 95% CI: 51.1–58.5) Canadian adults aged 65 and older reported having received the vaccine, with 60% of females and 48% of males vaccinated, falling short of the national target of 80% coverage [16]. To better characterize self-reported pneumococcal vaccine coverage and explore differences in vaccination rates among eligible Canadian adults – i.e., those aged 65 and older, as well as adults under 65 with at least one chronic medical condition (CMC) – using a large, national cohort study, we previously conducted a cross-sectional analysis of data collected from Canadian Longitudinal Study on Aging (CLSA) participants during the first CLSA follow-up conducted between 2015 and 2018 [17]. Our findings revealed that high proportions of eligible adults had not been vaccinated; 45.8% (95% CI: 45.2–46.5) of 22,246 CLSA participants aged 65 years and older and 81.3% (95% CI: 80.5–82.0) of 10,815 participants aged 47–64 years with at least one CMC reported never having received pneumococcal vaccine [17].

Recently, the CLSA released new data reporting pneumococcal vaccination status among a subset of participants collected during the follow-up 2 (FUP2) conducted from 2018–2021. We have conducted new cross-sectional analyses for the 2018–2021 period and achieve two objectives. First, we aimed to provide estimates of pneumococcal vaccine coverage among Canadian adults eligible for pneumococcal vaccination in the period 2018–2021 (FUP2) and quantify changes compared to coverage reported during 2015–2018 (FUP1). Our second objective was to identify sociodemographic, clinical, and healthcare utilization factors associated with being newly vaccinated, specifically among those participants who reported having been vaccinated when surveyed during the 2018–2021 FUP2 survey but had previously reported not having received any pneumococcal vaccine during the 2015–2018 FUP1 survey, despite being eligible at both time points.

## Methods

### Study setting and population

The CLSA is a national longitudinal cohort study initiated with 51,338 Canadian residents from all ten provinces aged between 45 and 85 years at the time of enrollment, which took place from 2011 to 2015 [18,19]. The CLSA comprises both the Comprehensive and Tracking cohorts, to which participants were recruited utilizing three distinct sampling frames and approaches [20,21]. Longitudinal cohort participants enrolled in the CLSA during the baseline data collection period, which took place from 2011 to 2015. Subsequently, the same participants were invited to complete an additional round of data collection approximately every three years during CLSA follow-up data collection. The initial follow-up survey (FUP1) was conducted from 2015–2018. The next follow up survey (FUP2) was conducted from 2018–2021. All questionnaires used within the CLSA are publicly available for review on the CLSA website [22]. The CLSA received ethical clearance from the McMaster University Health Integrated Research Ethics Board, as well as from corresponding research ethics boards at all collaborating Canadian institutions, to enroll participants into the cohort and collect data. CLSA participants authorized the use of their de-identified data for research purposes when they provided their consent prior to data collection. In this

manuscript, we report secondary analyses of previously collected, de-identified CLSA data accessed via an approved CLSA data request application. This data request and these analyses were approved by the Research Ethics Boards of McGill University (A02-E03-21A) and the University of Ottawa (H-01-23-8885).

## Data sources

We utilized the data collected from cohort participants during three distinct periods: the CLSA baseline study visit (conducted between 2011–2015), the CLSA FUP1 visit (conducted between 2015–2018), and the CLSA FUP2 visit (conducted between 2018–2021). Since in FUP2, vaccine-related queries were only asked to participants enrolled in the Comprehensive cohort, only participants from this cohort were included in the analysis. All survey questions and resulting variables used in our analyses are outlined in S1 Table to ensure reproducibility. For the purpose of the analyses presented here, data were accessed between 22/08/2023 and 30/04/2025.

## Outcome variable

During the FUP1 survey (2015–2018), participants from both the Comprehensive and the Tracking cohorts were asked to self-report their pneumococcal vaccination status by answering the question "have you had a pneumonia shot (pneumococcal vaccination) in your life?". At FUP2 (2018–2021), participants from the Comprehensive Cohort were asked the same question. For our first research aim, the outcome of interest was the "proportion that reported receiving pneumococcal vaccine", and participants were stratified as either vaccinated if they responded with a "yes," or unvaccinated if their response was "no." For our second aim, the outcome of interest was the "proportion that newly reported receiving pneumococcal vaccine" among CLSA participants.

## Eligibility criteria

Participants were eligible to be included in the analysis for our first aim if they 1) were asked the abovementioned question about their pneumococcal vaccination status at FUP2 and 2) were eligible to receive pneumococcal vaccine at the time of the survey, i.e., either aged ≥65 years or younger than 65 years old and reporting at least one underlying chronic medical condition (CMC). Participants were eligible to be included in the analysis for our second aim if they 1) were asked the abovementioned question about their pneumococcal vaccination status at both FUP1 and FUP2, 2) were eligible to receive pneumococcal vaccine at the time of both surveys (defined as previously indicated), and 3) reported not having received the vaccine at FUP1.

S1 Fig shows eligibility criteria for our analysis for both research aims.

## Sociodemographic variables

To ensure consistency and comparability to our previously published 2015–2018 analyses of pneumococcal vaccine coverage among CLSA participants [17], this analysis plan incorporated the following sociodemographic factors: sex at birth (male or female), age group (54 years or younger, 55–64, 65–74, 75–84, 85 years and older), racialized status (yes if non-white, no if white), highest education level (less than secondary school graduation, secondary school graduation without post-secondary education, some post-secondary education, post-secondary degree/diploma), annual household income (in Canadian dollars: < $20,000, $20,000 to <$50,000, $50,000 to <$100,000, $100,000 to <$150,000, $150,000 or higher), marital/partner status (single/never married/never lived with a partner, married or living with a partner in a common-law relationship, widowed, divorced/separated), province of residence (all Canadian provinces except for New Brunswick, Prince Edward Island and Saskatchewan as the CLSA Comprehensive Cohort does not include participants residing in one of these three provinces), urbanicity of residence (urban or rural). All the variables mentioned above were drawn from the FUP2 dataset, with the exception of sex at birth, racialized status, and education level which were only available from the baseline dataset.

## Variables related to health status and healthcare utilization

We categorized participants based on the presence or absence of at least one CMC, given that pneumococcal vaccination is recommended for adults with CMCs in Canada. Specifically, participants were queried about physician diagnoses pertaining to eight distinct groups of conditions. Each condition was dichotomously categorized as reported versus not reported diagnosis. These conditions included: cardiovascular disease (i.e., prior heart attack/myocardial infarction, angina or chest pain due to heart disease, hypertension), chronic lung disease (i.e., emphysema, chronic bronchitis, chronic obstructive pulmonary disease, chronic changes in lungs due to smoking, asthma), cerebrovascular disease (i.e., stroke and transient ischemic attack), chronic kidney disease or failure, diabetes mellitus, cancer, and chronic neurologic condition (i.e., dementia or Alzheimer's disease, Parkinsonism or Parkinson's disease, multiple sclerosis). These CMCs were included in this analysis because they are most likely to be associated with increased susceptibility to IPD among older adults and represent the majority of conditions outlined by the National Advisory Committee on Immunization (NACI) as eligibility criteria for pneumococcal vaccination in their recommendations [23]. This categorization aligns with our previous work regarding pneumococcal vaccination among CLSA participants, as well [17]. Additionally, we considered whether the participants' CMC status changed between FUP1 and FU2 (coded as 1 if one or more CMC were reported at FUP2 but not at FUP1, 0 if no changes were reported based on data from both timepoints).

To gauge healthcare utilization, we examined participants' self-reports on having any interactions with a family doctor within the 12 months preceding the FUP2 survey. Regarding participant's vaccination history, we evaluated receipt of an influenza vaccine in the previous 12 months, as self-reported at FUP2.

## Sample size and missing data

To assess pneumococcal vaccine coverage at FUP2, among 25,448 CLSA participants who completed the FUP2 survey, we excluded 23 participants whose province of residence changed between FUP1 and FUP2, moving to Prince Edward Island, Saskatchewan, or New Brunswick, i.e., provinces not covered by the Comprehensive Cohort. This decision was made to ensure data consistency and focus our analysis on provinces included in the Comprehensive Cohort. Moreover, we excluded 485 participants with no age available at the data collection site (DCS) visit at FUP2. In addition to this, birthdates were not available for secondary analysis as they are considered identifiable information; therefore, eligibility for pneumococcal vaccination could not be determined. Another 4.5% of participants from each of the two groups of interest (i.e., individuals aged 65 years and above and those aged <65 years with at least 1 CMC) were also excluded due to lack of outcome data (i.e., those who did not respond "yes" or "no" to the question on pneumococcal vaccination). Our final analysis sample included 10,530 adults aged ≥65 years and 4,055 adults <65 years with at least one CMC (S1 Fig).

Overall, the frequency of missing data among those included in the analysis was found to be lower than 1% for all sociodemographic, health status, and healthcare utilization variables except for income level at FUP2 (9.2% for adults aged ≥65 and 5.3% for adults aged <65 with at least one CMC).

To determine the proportion of newly reported vaccinated for pneumococcal disease at FUP2 relative to FUP1 (aim 2), we restricted our analyses to CLSA participants who were eligible to receive pneumococcal vaccine at both FUP1 and FUP2, and who reported not being vaccinated at FUP1 (i.e., 3,733 adults aged ≥65 and 2,776 adults aged <65 with at least one CMC). For the analysis examining factors associated with reporting being "newly vaccinated," we adopted a casewise deletion approach whenever we found missing data for any of the sociodemographic, health status, and healthcare utilization variables included in the models. The amount of missing data among participants with available outcome data was<1% for all variables.

## Statistical analysis

To assess the pneumococcal vaccine coverage, we calculated the proportion of participants who reported having received a pneumococcal vaccine among those eligible to receive pneumococcal vaccine (i.e., adults aged ≥65 years and adults

aged <65 years with at least one CMC). Proportions of individuals reporting being vaccinated and unvaccinated within each group were also calculated across strata defined by sociodemographic characteristics, CMCs, contact with a family doctor in the previous 12 months, and influenza vaccination in the previous 12 months. All 95% confidence intervals (CIs) were calculated through logit transformation of proportions.

To assess the proportion of newly vaccinated individuals, we used descriptive analyses to examine the characteristics of individuals who first reported being vaccinated for pneumococcal disease at FUP2 (newly reported vaccination) compared to those who reported remaining unvaccinated at both time points. To identify factors associated with newly reported vaccination, we used logistic regression models and included multiple covariates that have been shown to play a role in vaccination willingness and coverage. Specifically, our model for adults aged ≥65 years included age group, sex at birth, being racialized, highest education level, annual household income, marital/partner status, province of residence, urbanicity of residence, receipt of influenza vaccination in the 12 months prior to FUP2, reporting having been diagnosed with at least one CMC as of FUP2, and change in CMC status relative to FUP1. The model for adults <65 years with CMCs included the same sociodemographic factors as outlined above along with receipt of influenza vaccination in the previous 12 months. For each model, we reported adjusted odds ratios (aORs) and 95% CIs for the association between each independent variable and the outcome. Multicollinearity among predictors was assessed using variance inflation factors (VIF), indicating low collinearity and stable regression estimates. All analyses were conducted using the survey data commands in Stata version 18.0 (Stata-Corp, College Station, TX, USA) [24].

### Sensitivity analyses

As noted above, in FUP2, vaccine-related queries were only posed to individuals enrolled in the Comprehensive cohort, and since our previously published CLSA-based analysis using FUP1 data included both the Tracking and the Comprehensive cohort [17], we provide FUP1 analyses restricted to the Comprehensive cohort in the appendix; this is meant for comparative purposes, though it is important to highlight that no meaningful differences were observed at FUP1 between the overall cohort and the Comprehensive cohort alone. We adopted the same rationale and approach to analyse pneumococcal vaccine coverage among Comprehensive cohort participants at FUP1, whose results are reported in the supplementary material for comparative purposes.

The proportion of CLSA participants who were asked questions about pneumococcal vaccination was lower at FUP2 compared to FUP1. This decrease resulted from an administrative decision during the implementation of the FUP2 survey, which temporarily removed the Preventive Health Behaviours (PHB) module, including the pneumococcal vaccination questions from the survey between early 2018 and early 2019. To determine if there were systematic differences between those who were asked the question at FUP2 and those who were not, we compared the distribution of selected sociodemographic variables between the two groups.

Additionally, since 9.2% of adults aged 65 and older, and 5.3% of those under 65 with at least one chronic medical condition (CMC) were missing income data at FUP2, we conducted a sensitivity analysis by imputing the missing income values at FUP2 using the income levels reported at FUP1.

### Results

#### Prevalence of reported pneumococcal vaccination among CLSA participants based on Follow up 1 (2018–2021)

The sociodemographic characteristics of CLSA participants aged 65 and older and those aged 49–64 years with one or more CMCs, stratified by pneumococcal vaccination status reported at FUP2 (2018–2021), are presented in Table 1. For comparison, the sociodemographic characteristics of participants at FUP1 (2015–2018) restricted to the Comprehensive cohort are provided in S2 Table.

**Table 1. Self-reported pneumococcal vaccination status among Canadian Longitudinal Study on Aging (CLSA) participants who were eligible to receive a pneumococcal vaccine as per Canada's National Advisory Committee on Immunization (NACI) guidelines, by key sociodemographic characteristics at follow-up 2 (FUP2; 2018-2021).**

| Characteristic | Self-reported pneumococcal vaccination status in lifetime | | | | | | | |
| --- | --- | --- | --- | --- | --- | --- | --- | --- |
| | Individuals aged 65 and older (N=10,530) | | | | Individuals aged <65 with at least one CMC (N=4,055) | | | |
| | Vaccinated | | Unvaccinated | | Vaccinated | | Unvaccinated | |
| | N | % (95% CI) | N | % (95% CI) | N | % (95% CI) | N | % (95% CI) |
| Overall | 5976 | 56.8 (55.8-57.7) | 4554 | 43.2 (42.3-44.2) | 781 | 19.3 (18.1-20.5) | 3274 | 80.7 (79.5-81.9) |
| **Sex at birth** | | | | | | | | |
| Female | 3250 | 62.0 (60.7-63.4) | 1988 | 38.0 (36.6-39.3) | 424 | 21.0 (19.3-22.8) | 1597 | 79.0 (77.2-80.7) |
| Male | 2726 | 51.5 (50.2-52.9) | 2566 | 48.5 (47.1-49.8) | 357 | 17.6 (16.0-19.3) | 1677 | 82.4 (80.7-84.0) |
| Missing | 0 | 0.0 (N/A) | 0 | 0.0 (N/A) | 0 | 0.0 (N/A) | 0 | 0.0 (N/A) |
| **Age group** | | | | | | | | |
| <55 | N/A | N/A | N/A | N/A | 79 | 13.0 (10.6-15.9) | 528 | 87.0 (84.1-89.4) |
| 55-64 | N/A | N/A | N/A | N/A | 702 | 20.4 (19.0-21.7) | 2746 | 79.6 (78.3-81.0) |
| 65-74 | 3005 | 51.6 (50.3-52.8) | 2824 | 48.4 (47.2-49.7) | N/A | N/A | N/A | N/A |
| 75-84 | 2182 | 62.6 (61.0-64.2) | 1304 | 37.4 (35.8-39.0) | N/A | N/A | N/A | N/A |
| 85+ | 789 | 64.9 (62.2-67.6) | 426 | 35.1 (32.4-37.8) | N/A | N/A | N/A | N/A |
| **Racialized** | | | | | | | | |
| No | 5756 | 57.2 (56.2-58.1) | 4313 | 42.8 (41.9-43.8) | 730 | 19.4 (18.1-20.7) | 3037 | 80.6 (79.3-81.9) |
| Yes | 214 | 47.7 (43.1-52.3) | 235 | 52.3 (47.7-56.9) | 49 | 17.1 (13.2-22.0) | 237 | 82.9 (78.0-86.8) |
| Missing | 6 | 50.0 (24.4-75.6) | 6 | 50.0 (24.4-75.6) | 02 | 100 (N/A) | 0 | 0.0 (N/A) |
| **Higher education level** | | | | | | | | |
| Less than second. school educ. | 357 | 56.0 (52.2-59.9) | 280 | 44.0 (40.1-47.8) | 24 | 24.7 (17.2-34.3) | 73 | 75.3 (65.7-82.8) |
| Second. school grad., no post-second. school educ. | 577 | 57.6 (54.3-60.4) | 429 | 42.4 (39.6-45.7) | 62 | 21.1 (16.8-26.1) | 232 | 78.9 (73.9-23.2) |
| Some post-second. educ. | 456 | 52.7 (49.2-55.8) | 412 | 47.5 (44.2-50.8) | 65 | 22 (17.7-27.1) | 230 | 78 (72.9-82.3) |
| Post-second. degree/diploma | 4575 | 57.2 (56.1-58.3) | 3420 | 42.8 (41.7-43.9) | 630 | 18.7 (17.4-20.1) | 2738 | 81.3 (79.9-82.6) |
| Missing | 11 | 45.8 (27.5-65.4) | 13 | 54.2 (34.6-72.5) | 0 | 0.0 (N/A) | 01 | 100 (N/A) |
| **Annual household income (in Canadian dollars)** | | | | | | | | |
| Less than $20,000 | 285 | 52.6 (48.4-56.8) | 257 | 47.4 (43.2-51.6) | 42 | 26.8 (20.4-34.2) | 115 | 73.2 (65.8-79.6) |
| $20,000 to <$50,000 | 1365 | 53.9 (51.9-55.8) | 1169 | 46.1 (44.2-48.1) | 96 | 20.9 (17.4-24.8) | 364 | 79.1 (75.2-82.6) |
| $50,000 to <$100,000 | 2306 | 59.1 (57.6-60.6) | 1595 | 40.9 (39.4-42.4) | 257 | 21.9 (19.7-24.4) | 914 | 78.1 (75.6-80.3) |
| $100,000 to <$150,000 | 942 | 57.4 (55.0-59.8) | 698 | 42.6 (40.2-45.0) | 169 | 17.9 (15.6-20.4) | 777 | 82.4 (79.6-84.4) |
| $150,000 or higher | 533 | 56.2 (53.0-59.4) | 415 | 43.8 (40.6-47.0) | 161 | 14.6 (12.6-16.8) | 945 | 85.4 (83.2-87.4) |
| Missing | 545 | 56.2 (53.0-59.4) | 420 | 43.8 (40.6-47.0) | 56 | 26.0 (20.6-32.3) | 159 | 74.0 (67.7-79.4) |
| **Marital/partner status** | | | | | | | | |
| Single/never married/never lived with a partner | 412 | 52.6 (49.0-56.0) | 372 | 47.4 (44.0-51.0) | 116 | 25.2 (21.5-29.4) | 344 | 74.8 (70.6-78.5) |
| Married/Common-law | 3659 | 56.3 (55.1-57.5) | 2844 | 43.7 (42.5-44.9) | 529 | 18.2 (16.8-34.3) | 2376 | 74.4 (65.7-81.5) |
| Widowed | 1165 | 64.0 (61.8-66.2) | 654 | 36.0 (33.8-38.2) | 30 | 25.6 (18.5-34.3) | 87 | 74.4 (65.7-81.5) |
| Divorced/Separated | 712 | 51.7 (49.1-54.4) | 664 | 48.3 (45.6-50.9) | 103 | 18.8 (15.7-22.3) | 445 | 81.2 (77.7-84.3) |
| Missing | 28 | 58.3 (44.1-71.3) | 20 | 41.7 (28.7-55.9) | 3 | 12.0 (3.9-31.3) | 22 | 88.0 (68.7-96.1) |
| **Province of residence** | | | | | | | | |
| Newfoundland | 235 | 35.6 (32.0-39.3) | 425 | 64.4 (60.7-68.0) | 30 | 10.8 (7.6-15.0) | 249 | 89.2 (85.0-92.4) |
| Nova Scotia | 534 | 52.9 (49.8-55.9) | 476 | 47.1 (44.1-50.2) | 79 | 20.7 (16.9-25.0) | 303 | 79.3 (75.0-83.1) |
| Quebec | 1323 | 60.1 (58.0-62.1) | 879 | 39.9 (37.9-42.0) | 154 | 19.1 (16.5-21.9) | 653 | 80.9 (78.1-83.5) |
| Ontario | 1362 | 58.3 (56.2-60.2) | 976 | 41.7 (39.8-43.8) | 190 | 21.4 (18.8-24.2) | 699 | 78.6 (75.8-81.2) |
| Manitoba | 586 | 60.7 (57.6-63.8) | 379 | 39.3 (36.2-42.4) | 62 | 16.4 (13.0-20.4) | 317 | 83.6 (79.6-87.0) |

*(Continued)*

**Table 1.** (Continued)

| Characteristic | Self-reported pneumococcal vaccination status in lifetime | | | | | | | |
| | Individuals aged 65 and older (N = 10,530) | | | | Individuals aged <65 with at least one CMC (N = 4,055) | | | |
| | Vaccinated | | Unvaccinated | | Vaccinated | | Unvaccinated | |
| | N | % (95% CI) | N | % (95% CI) | N | % (95% CI) | N | % (95% CI) |
| Alberta | 718 | 65.9 (63.1-68.7) | 371 | 34.1 (31.3-36.9) | 89 | 21.4 (13.0-20.4) | 326 | 78.6 (74.3-82.2) |
| British Columbia | 1218 | 53.8 (51.7-55.8) | 1048 | 46.2 (44.2-48.3) | 177 | 19.6 (17.1-22.3) | 727 | 80.4 (77.7-82.9) |
| **Urbanicity of residence** | | | | | | | | |
| Urban | 5558 | 57.2 (56.2-58.2) | 4157 | 42.8 (41.8-43.8) | 722 | 19.6 (18.4-20.9) | 2958 | 80.4 (79.1-81.6) |
| Rural | 400 | 51.5 (48.0-55.0) | 377 | 48.5 (45.0-52.0) | 53 | 14.8 (11.5-18.9) | 305 | 85.2 (81.1-88.5) |
| Missing | 18 | 47.4 (32.3-63.0) | 20 | 52.6 (37.0-67.7) | 6 | 35.3 (16.8-59.6) | 11 | 64.7 (40.4-83.2) |
| **Chronic Medical Condition status** | | | | | | | | |
| None reported | 1107 | 46.5 (44.5-48.5) | 1272 | 53.5 (51.5-55.5) | 270 | 7.9 (7.1-8.9) | 3137 | 92.1 (91.1-92.9) |
| At least one reported | 4830 | 59.7 (58.7-60.8) | 3254 | 40.3 (39.2-53.8) | 781 | 19.3 (18.1-20.5) | 3274 | 80.7 (79.5-81.9) |
| Missing | 39 | 58.2 (46.2-69.4) | 28 | 41.8 (30.6-53.8) | 6 | 10.2 (4.6-20.8) | 53 | 89.8 (79.2-95.4) |
| **Contact with family doctor in previous 12 months** | | | | | | | | |
| No | 340 | 39.5 (36.3-42.8) | 520 | 60.5 (57.2-63.7) | 44 | 10.3 (7.8-13.6) | 382 | 89.7 (86.4-92.2) |
| Yes | 5630 | 58.3 (57.3-59.3) | 4027 | 41.7 (40.7-42.7) | 737 | 20.3 (19.1-21.7) | 2888 | 79.7 (78.3-80.9) |
| Missing | 6 | 46.2 (22.4-71.8) | 7 | 53.8 (28.2-77.6) | 0 | 0.0 (N/A) | 4 | 100 (N/A) |
| **Receipt of influenza vaccine in previous 12 months (self-reported)** | | | | | | | | |
| No | 727 | 25.5 (23.9-27.1) | 2125 | 74.5 (72.9-76.1) | 155 | 8.4 (7.2-9.8) | 1688 | 91.6 (90.2-92.8) |
| Yes | 5241 | 68.4 (67.4-69.5) | 2417 | 31.6 (30.5-32.6) | 625 | 28.3 (26.5-30.2) | 1582 | 71.7 (69.8-73.5) |
| Missing | 8 | 40.0 (21.4-62.0) | 12 | 60.0 (38.0-78.6) | 1 | 20.0 (2.7-69.1) | 4 | 80.0 (30.9-97.3) |

CMC, Chronic Medical Condition.

Among 10,530 participants aged 65 and older with information on self-reported pneumococcal vaccination status at FUP2, 43.2% (95% CI: 42.3–44.2) reported not being vaccinated. The proportion of non-vaccinated participants was higher among males than females, in the 65–74 age group versus older age groups, among racialized versus non-racialized participants, and in certain provinces (Newfoundland, Nova Scotia, and British Columbia) compared to others and among those who resided in rural versus urban areas. At FUP2 (2018–2021), 76.7% of participants aged 65 and older reported having been diagnosed with at least one CMC. Among these individuals, the proportion of non-vaccinated was slightly lower (40.3% [95% CI: 39.2–53.8]) compared to participants without any CMCs (53.5% [95% CI 51.5–55.5]). The distribution of CMCs by self-reported pneumococcal vaccination status among those 65 years and older at both FUP1 (comprehensive cohort) and FUP2 is provided in S3 and S4 Tables, respectively. A higher proportion of participants reported not being vaccinated among those who reported not having had contact with a family doctor in the previous 12 months leading up to FUP2 versus those who did have contact. Furthermore, 74.5% (95% CI: 72.9–76.1) of participants who reported not having received the influenza vaccine in the previous 12 months also reported not being vaccinated for pneumococcal disease (Table 1).

Among 4,055 participants aged 49–64 with one or more CMCs, 80.7% (95% CI: 79.5–81.9) reported not having received a pneumococcal vaccine at FUP2 (Table 1). In this group, a higher proportion of those in the younger age group (<55 years), higher income categories, and residents in Newfoundland versus other provinces reported not receiving pneumococcal vaccine. Proportions of vaccinated individuals were found to be comparable between sexes, in racialized versus non-racialized subjects, and between urban and rural residents. The distribution of CMCs by self-reported

pneumococcal vaccination status for FUP1 and FUP2 is available in S3 and S4 Tables, respectively. As with older adults, non-vaccination rates among participants aged 47–64 with CMCs were higher for those who reported no contact with a family doctor in the previous year (89.7% [95% CI: 86.4–92.2)]. Additionally, 91.6% (95% CI: 90.2–92.8) of those who reported not having received the influenza vaccine in the previous 12 months also indicated they had not received the pneumococcal vaccine (Table 1).

## Factors associated with reporting being newly vaccinated for pneumococcal disease during the 2018–2021 survey

Among 3,733 participants aged 65 and older at FUP2 who reported not having received a pneumococcal vaccine at FUP1 despite being already age-eligible at that time, 71.6% (95% CI: 70.1–73.0) reported still being unvaccinated at FUP2. Only 1,061 (28.4% [95% CI: 27.0–29.9]) first reported having received a pneumococcal vaccine at FUP2 (newly vaccinated). Similarly, among 2,776 participants aged 49–64 with one or more CMCs who were eligible for pneumococcal vaccination at both FUP1 and FUP2, only 11% (95% CI: 9.9–12.2) first reported being vaccinated at FUP2, leaving 89.0% [95% CI: 87.8–90.1] of participants still non-vaccinated (Table 2).

After adjusting for sociodemographic factors, several covariates were found to be associated with reporting being newly vaccinated for pneumococcal disease among participants aged 65 and older. With respect to the province of residence, Quebec showed the highest odds of new vaccination (aOR = 1.60 [95% CI: 1.20–2.11]), while Newfoundland had the lowest odds (aOR = 0.44 [95% CI: 0.32–0.61]), compared to Ontario. Males had lower odds of becoming newly vaccinated for pneumococcal disease compared to females (aOR = 0.61 [95% CI: 0.51–0.74]). Age was also associated with reporting new vaccination, with individuals aged 75–84 being less likely to report being newly vaccinated (aOR = 0.70 [95% CI: 0.58–0.85]) compared to those aged 65–74 (Fig 1).

Individuals who reported having received the influenza vaccine and had contact with a family doctor in the previous 12 months were significantly more likely to report being newly vaccinated with pneumococcal vaccine (aOR = 6.70 [95% CI: 5.40–8.32]) and (aOR = 1.94 [95% CI: 1.35–2.79]), respectively. Moreover, participants who first reported having been diagnosed with one of the CMCs investigated at FUP2 had higher odds of being newly vaccinated (aOR = 1.41 [95% CI: 1.01–1.96]) compared to those with no change in CMC status (Fig 1).

Among CLSA participants aged 49–64 with at least one CMC, similar factors were found to be associated with new pneumococcal vaccination. The odds of becoming newly vaccinated were lowest among those who reported having some post-secondary education (aOR = 0.33 [95% CI: 0.14–0.80]) and those with a post-secondary degree or diploma (aOR = 0.44 [95% CI: 0.23–0.87]) versus those with less than secondary education. Additionally, individuals residing in Newfoundland (aOR = 0.35 [95% CI: 0.18–0.68]) and Manitoba (aOR = 0.42 [95% CI: 0.23–0.78]) were less likely to report being newly vaccinated compared to those residing in Ontario. Similar to older adults, participants in this group who had received the influenza vaccine and had contact with family doctor in the previous 12 months were more likely to report being newly vaccinated for pneumococcal disease (aOR = 4.33 [95% CI: 3.17–5.92]) and (aOR = 2.10 [95% CI: 1.18–3.73]), respectively (Fig 2).

## Sensitivity analysis

We utilized data exclusively from the Comprehensive cohort at FUP1 (2015–2018) to apply the same eligibility criteria as in our FUP2 analysis and better compare pneumococcal vaccination status between the two timepoints. Among the 13,366 eligible participants aged 65 and older who provided self-reported information about their pneumococcal vaccination status at FUP1, 46.3% (95% CI: 45.4–47.1) reported not being vaccinated. This reflects a slight increase compared to FUP2 (S2 Table). Among the 6,714 participants aged 49–64 with one or more chronic medical conditions (CMCs), 82.8% (95% CI: 81.8–83.7) indicated they were not vaccinated (S2 Table).

**Table 2. Characteristics of Canadian Longitudinal Study on Aging (CLSA) participants who reported being newly vaccinated for pneumococcal disease at follow-up 2 (FUP2) since FUP1, versus those still non-vaccinated. Only participants who were considered vaccine-eligible at both FUP1 and FUP2 and who reported not being vaccinated at FUP1 were included in this analysis.**

| Characteristic | Self-reported pneumococcal vaccination status in lifetime | | | | | | | |
|---|---|---|---|---|---|---|---|---|
| | Individuals aged 65 and older (N=3733) | | | | Individuals aged <65 with at least 1 CMC (N=2776) | | | |
| | Still Unvaccinated | | Newly Unvaccinated | | Still Unvaccinated | | Newly Vaccinated | |
| | N | % (95% CI) | N | % (95% CI) | N | % (95% CI) | N | % (95% CI) |
| Overall | 2672 | 71.6 (70.1-73.0) | 1061 | 28.42 (27.0-29.9) | 2470 | 89.0 (87.8-90.1) | 306 | 11.0 (9.9-12.2) |
| **Sex at birth** | | | | | | | | |
| Female | 1218 | 69.5 (67.3-71.6) | 535 | 30.5 (28.4-32.7) | 1285 | 89.2 (87.5-90.7) | 156 | 10.8 (9.3-12.5) |
| Male | 1454 | 73.4 (71.4-75.3) | 526 | 26.6 (24.7-28.6) | 1185 | 88.8 (87.0-90.4) | 150 | 11.2 (9.6-13.0) |
| **Age group** | | | | | | | | |
| <55 | N/A | N/A | N/A | N/A | 398 | 91.3 (88.2-93.6) | 38 | 8.7 (6.4-11.8) |
| 55-64 | N/A | N/A | N/A | N/A | 2072 | 88.5 (87.2-89.8) | 268 | 11.5 (10.2-12.8) |
| 65-74 | 1419 | 68.8 (66.7-70.7) | 645 | 31.3 (29.3-33.3) | N/A | N/A | N/A | N/A |
| 75-84 | 978 | 74.5 (72.1-76.8) | 334 | 25.5 (23.2-27.9) | N/A | N/A | N/A | N/A |
| 85+ | 275 | 77.0 (72.4-81.1) | 82 | 23.0 (18.9-27.6) | N/A | N/A | N/A | N/A |
| **Racialized** | | | | | | | | |
| No | 2534 | 71.5 (70.0-72.9) | 1011 | 28.5 (27.1-30.0) | 2295 | 89.0 (87.8-90.2) | 283 | 11.0 (9.8-12.2) |
| Yes | 134 | 73.2 (66.3-79.1) | 49 | 26.8 (20.9-33.7) | 175 | 88.8 (83.6-92.5) | 22 | 11.2 (7.5-16.4) |
| Missing | 4 | 80.0 (30.9-97.3) | 1 | 20.0 (2.7-69.1) | 0 | 0.0 (N/A) | 1 | 100.0 (N/A) |
| **Highest education level** | | | | | | | | |
| Less than second. school educ. | 191 | 76.1 (70.4-81.0) | 60 | 23.9 (19.0-29.6) | 57 | 81.4 (70.6-88.9) | 13 | 18.6 (11.1-29.4) |
| Second. school grad., no post-second. school educ. | 251 | 71.5 (66.6-76.0) | 100 | 28.5 (24.0-33.4) | 179 | 86.9 (81.6-90.9) | 27 | 13.1 (9.1-18.4) |
| Some post-second. educ. | 230 | 74.9 (69.8-79.5) | 77 | 25.1 (20.5-30.2) | 178 | 90.4 (85.4-93.8) | 19 | 9.6 (6.2-14.6) |
| Post-second. degree/diploma | 1989 | 70.8 (69.0-72.4) | 822 | 29.2 (27.6-31.0) | 2056 | 89.3 (87.9-90.5) | 247 | 10.7 (9.5-12.1) |
| Missing | 11 | 84.6 (54.9-96.1) | 2 | 15.4 (3.9-45.1) | 0 | N/A | 0 | N/A |
| **Annual household income (in Canadian dollars)** | | | | | | | | |
| Less than $20,000 | 162 | 76.8 (70.6-82.0) | 49 | 23.2 (18.0-29.4) | 88 | 88.9 (81.0-93.7) | 11 | 11.1 (6.3-19.0) |
| $20,000 to <$50,000 | 783 | 77.4 (74.8-79.9) | 228 | 22.6 (20.1-25.2) | 282 | 87.6 (83.5-90.8) | 40 | 12.4 (9.2-16.5) |
| $50,000 to <$100,000 | 937 | 68.1 (65.6-70.6) | 438 | 31.9 (29.4-34.4) | 702 | 88.5 (86.1-90.6) | 91 | 11.5 (9.4-13.9) |
| $100,000 to <$150,000 | 357 | 67.5 (63.4-71.3) | 172 | 32.5 (28.7-36.6) | 591 | 89.5 (87.0-91.7) | 69 | 10.5 (8.3-13.0) |
| $150,000 or higher | 181 | 68.0 (62.2-73.4) | 85 | 32.0 (26.6-37.8) | 685 | 90.6 (88.3-92.5) | 71 | 9.4 (7.5-11.7) |
| Missing | 252 | 73.9 (69.0-78.3) | 89 | 26.1 (21.7-31.0) | 122 | 83.6 (76.6-88.7) | 24 | 16.4 (11.3-23.4) |
| **Marital/partner status** | | | | | | | | |
| Single/never married/never lived with a partner | 187 | 70.6 (64.8-75.7) | 78 | 29.4 (24.3-35.2) | 269 | 87.3 (83.1-90.6) | 39 | 12.7 (9.4-16.9) |
| Married/Common-law | 1613 | 70.2 (68.3-72.0) | 686 | 29.8 (28.0-31.7) | 1786 | 88.8 (87.4-90.1) | 225 | 11.2 (9.9-12.6) |
| Widowed | 457 | 74.8 (71.2-78.1) | 154 | 25.2 (21.9-28.8) | 65 | 90.3 (81.0-95.3) | 7 | 9.7 (4.7-19.0) |
| Divorced/Separated | 413 | 74.3 (70.5-77.7) | 143 | 25.7 (22.3-29.5) | 348 | 90.9 (87.5-93.4) | 35 | 9.1 (6.6-12.5) |
| Missing | 2 | 100.0 (N/A) | 0 | 0.0 (N/A) | 2 | 100.0 (N/A) | 0 | 0.0 (N.A) |
| **Province of residence** | | | | | | | | |
| Newfoundland | 304 | 78.4 (74.0-82.2) | 84 | 21.6 (17.8-26.0) | 210 | 94.6 (90.7-96.9) | 12 | 5.4 (3.1-9.3) |
| Nova Scotia | 276 | 76.5 (71.8-80.5) | 85 | 23.5 (19.5-28.2) | 169 | 88.5 (83.1-92.3) | 22 | 11.5 (7.7-16.9) |
| Quebec | 543 | 73.6 (70.3-76.6) | 195 | 26.4 (23.4-29.7) | 508 | 89.0 (86.1-91.3) | 63 | 11.0 (8.7-13.9) |
| Ontario | 543 | 67.0 (63.7-70.2) | 267 | 33.0 (29.8-36.3) | 546 | 86.3 (83.3-88.7) | 87 | 13.7 (11.3-16.7) |
| Manitoba | 204 | 71.3 (65.8-76.3) | 82 | 28.7 (23.7-34.2) | 228 | 93.4 (89.6-95.9) | 16 | 6.6 (4.1-10.4) |
| Alberta | 200 | 65.1 (59.6-70.3) | 107 | 34.9 (29.7-40.4) | 251 | 86.6 (82.1-90.0) | 39 | 13.4 (10.0-17.9) |

*(Continued)*

**Table 2.** (Continued)

| Characteristic | Self-reported pneumococcal vaccination status in lifetime | | | | | | | |
| --- | --- | --- | --- | --- | --- | --- | --- | --- |
| | Individuals aged 65 and older (N = 3733) | | | | Individuals aged <65 with at least 1 CMC (N = 2776) | | | |
| | Still Unvaccinated | | Newly Unvaccinated | | Still Unvaccinated | | Newly Vaccinated | |
| | N | % (95% CI) | N | % (95% CI) | N | % (95% CI) | N | % (95% CI) |
| British Columbia | 602 | 71.4 (68.3-73.0) | 241 | 28.4 (27.0-29.9) | 558 | 89.3 (86.6-91.5) | 67 | 10.7 (8.5-13.4) |
| **Urbanicity of residence** | | | | | | | | |
| Urban | 2438 | 71.3 (69.8-72.8) | 981 | 28.7 (27.2-30.2) | 2232 | 88.7 (87.5-89.9) | 283 | 11.3 (10.1-12.5) |
| Rural | 224 | 74.4 (69.2-79.0) | 77 | 25.6 (21.0-30.8) | 230 | 91.6 (87.5-94.5) | 21 | 8.4 (5.5-12.5) |
| Missing | 10 | 76.9 (47.8-92.4) | 3 | 23.1 (7.6-52.2) | 8 | 80.0 (45.9-95.0) | 2 | 20.0 (5.0-54.1) |
| **Chronic Medical Condition status** | | | | | | | | |
| None reported | 704 | 74.3 (71.5-77.0) | 243 | 25.7 (23.0-28.5) | 236 | 93.3 (89.5-95.8) | 17 | 6.7 (4.2-10.5) |
| At least one reported | 1951 | 70.6 (68.9-72.3) | 811 | 29.4 (27.7-31.1) | 2470 | 89.0 (87.8-90.1) | 306 | 11.0 (9.9-12.2) |
| Missing | 17 | 70.8 (50.2-85.4) | 7 | 29.2 (14.6-49.8) | 20 | 95.2 (72.8-99.3) | 1 | 4.8 (0.7-27.2) |
| **Contact with family doctor in previous 12 months** | | | | | | | | |
| No | 293 | 85.2 (81.0-88.6) | 51 | 14.8 (11.4-19.0) | 267 | 94.7 (91.4-96.8) | 15 | 5.3 (3.2-8.6) |
| Yes | 2374 | 70.2 (68.6-71.7) | 1009 | 29.8 (28.3-31.4) | 2200 | 88.3 (87.0-89.5) | 291 | 11.7 (10.5-13.0) |
| Missing | 5 | 83.3 (36.9-97.7) | 1 | 16.7 (2.3-63.1) | 3 | 100.0 (N/A) | 0 | 0.0 (N/A) |
| **Receipt of influenza vaccine in previous 12 months (self-reported)** | | | | | | | | |
| No | 1355 | 88.9 (87.2-90.3) | 170 | 11.1 (9.7-12.8) | 1295 | 94.8 (93.5-95.9) | 71 | 5.2 (4.1-6.5) |
| Yes | 1313 | 59.7 (57.6-61.7) | 888 | 40.3 (38.3-42.4) | 1172 | 83.4 (81.3-85.2) | 234 | 16.6 (14.8-18.7) |
| Missing | 4 | 57.1 (23.0-85.6) | 3 | 42.9 (14.4-77.0) | 3 | 75.0 (23.8-96.7) | 1 | 25.0 (3.3-76.2) |

CMC, Chronic medical condition.

To identify systematic differences between those who were asked about their vaccination status at FUP2 and those who were not, we compared the distribution of selected sociodemographic variables between the two groups (S5 Table). Notably, we observed some differences in sex and age distribution between the participants who were asked about pneumococcal vaccination and those who were not, as well as minor differences in distribution based on province of residence.

Given the amount of missing income data at FUP2, we investigated changes in reported income level between FUP1 and FUP2 and did not observe meaningful differences as illustrated in S2 Fig. In our sensitivity analyses for factors associated with pneumococcal vaccination for participants aged 49–64 with at least one CMC and participants aged 65 and older, we did not observe significant changes in adjusted model estimates obtained after imputing missing income data (S6 and S7 Tables).

## Discussion

Despite efforts to increase vaccination coverage, pneumococcal vaccine coverage among older adults at risk of IPD remains suboptimal in Canada. The CLSA cohort provides a unique opportunity to assess changes in pneumococcal vaccination coverage over time and to evaluate factors associated with recent pneumococcal vaccine coverage. Our analyses of the CLSA 2018–2021 (FUP2) data suggest only slight increases in pneumococcal vaccination rates compared to our previously published analysis of FUP1 (2015–2018) [17]. Among participants aged 65 and older, 43.2% remained unvaccinated for pneumococcal disease, reflecting only a minimal decrease in non-vaccination rates from our previously published FUP1 analysis [17] and the FUP1 analysis restricted to the comprehensive cohort reported here (46%). In the 49–64 age group with at least one comorbidity, 80.7% reported not being vaccinated, indicating a small improvement from

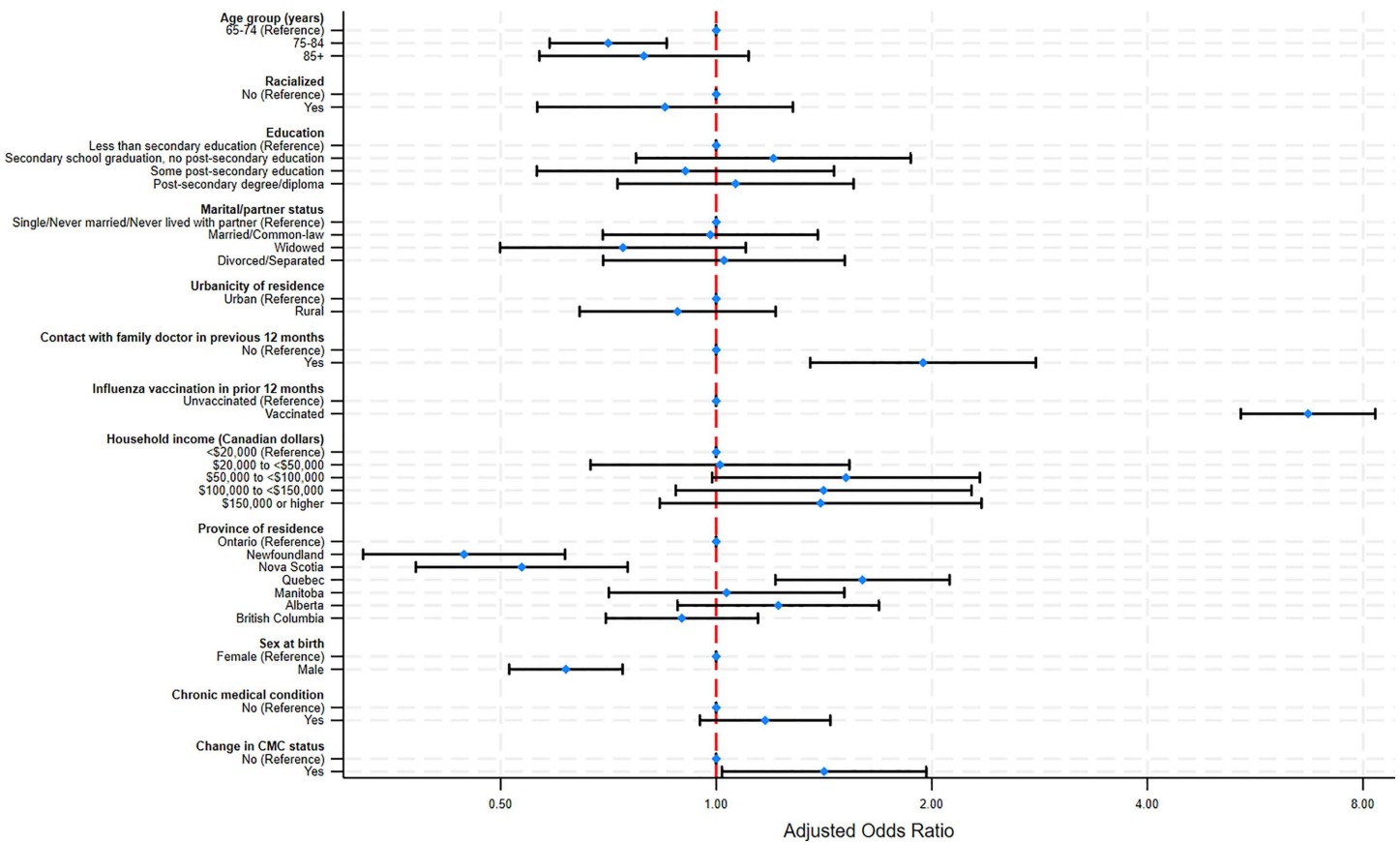

**Fig 1. Factors associated with changes in self-reported pneumococcal vaccination status among older adults (aged 65 and older).** Logistic regression analysis of Canadian Longitudinal Study on Aging (CLSA) participants aged 65 years and older who reported being newly vaccinated with pneumococcal vaccine during FUP2 (n = 3,134).

the 82% non-vaccination rate seen in FUP1 (both our previously published and the analysis restricted to the comprehensive cohort reported here). Similar to the findings from FUP1 (2015–2018) analyses, substantial non-vaccination rates were observed in specific demographic subgroups among participants aged 65 and older. Males, individuals aged 65–74, certain provincial groups, and rural residents were less likely to be vaccinated. Younger individuals (<55) and those from higher income brackets within the 49–64 age group with CMCs also exhibited high non-vaccination rates, indicating no significant change compared to our FUP1 estimates. Our study identified two key predictors of pneumococcal vaccination at FUP2 among those already eligible but not vaccinated at FUP1: receipt of influenza vaccination and recent contact with a family physician. Influenza vaccination is consistently associated with higher pneumococcal vaccination coverage [17,25], while regular engagement with primary care providers underscores the role of healthcare interactions in promoting and providing an opportunity for vaccination [26,15]. Sociodemographic patterns in vaccination behavior remained stable between FUP1 and FUP2. As part of the FUP2, data were collected during the early phases of the COVID-19 pandemic; the stagnant vaccination rates observed in this work could be potentially related to the disruption in vaccination services, the decreased utilization of healthcare services, and changes in vaccination behaviors related to the pandemic. However, further investigation is needed to determine if, and to what extent, disruption in vaccination services, decreased utilization of healthcare services, and changes in vaccination behaviors related to the COVID-19 pandemic may have influenced pneumococcal vaccination uptake among older adults during this period. Furthermore, the high proportion of participants

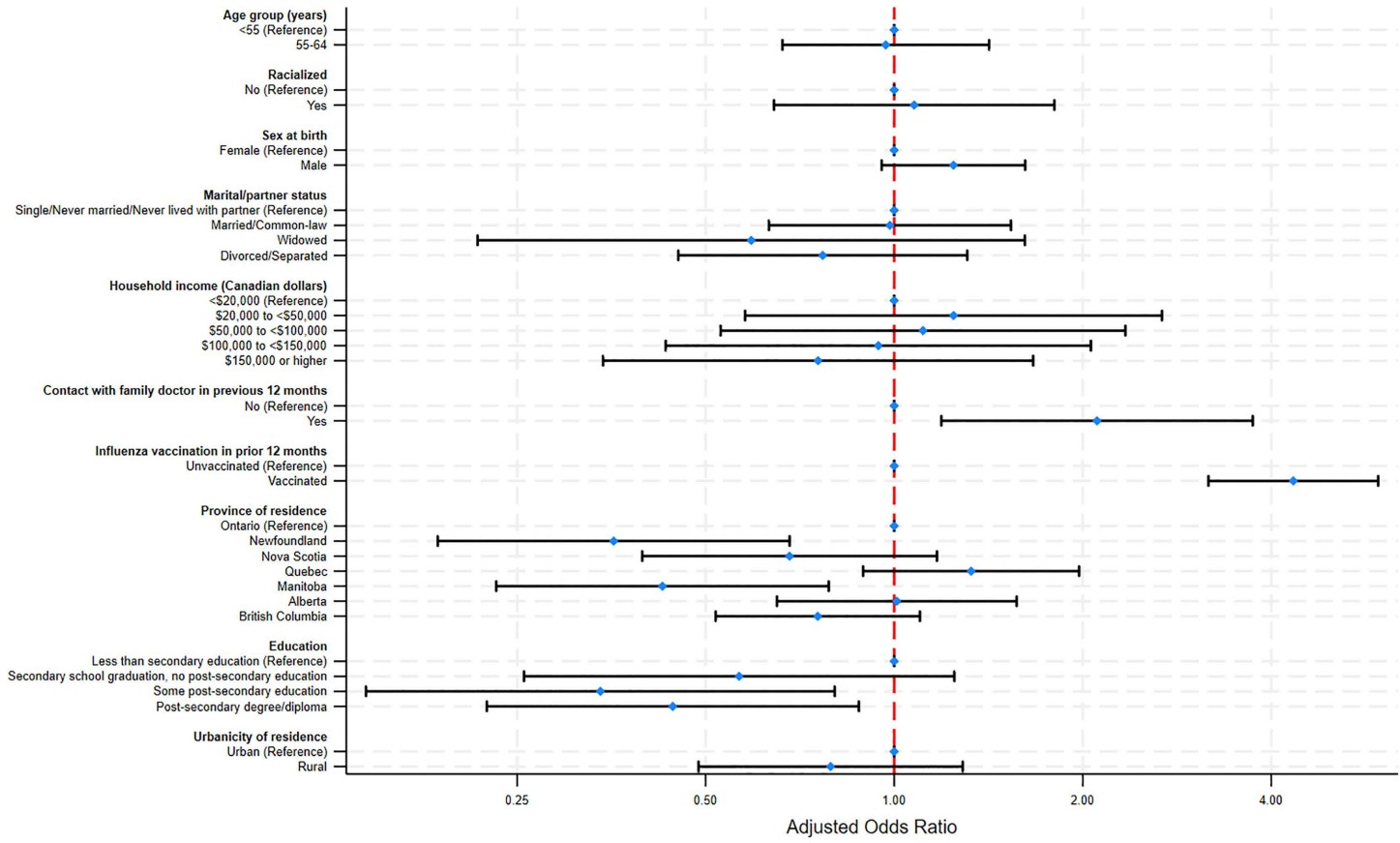

**Fig 2. Factors associated with changes in self-reported pneumococcal vaccination status among adults (55-64 years) with chronic medical conditions.** Logistic regression analysis of Canadian Longitudinal Study on Aging (CLSA) participants aged 49-64 years with one or more chronic medical conditions (CMCs) being newly vaccinated with pneumococcal vaccine during FUP2 (n = 2,611).

still unvaccinated for pneumococcal disease based on this analysis of the most recent data collected in 2018–2021 signals potential systemic barriers that warrant targeted, population-specific approaches to improve vaccination coverage rates.

Our findings indicate sex-associated disparities in self-reported pneumococcal vaccination status, with males consistently reporting lower vaccination rates compared to females. This observation aligns with other studies showing that females have a higher propensity for vaccination and often act as vaccination advocates within their families and communities [25,15,27]. Provincial differences in vaccine coverage were also noted, with participants from Newfoundland reporting being unvaccinated more frequently in both study groups, mirroring findings from the Adult National Immunization Survey 2023 [28] as well as our previous analysis assessing pneumococcal vaccination among CLSA participants during 2015–2018 [17]. Together with local, community-based insights, these insights can inform region-specific strategies to boost vaccination rates. Approaches such as enhancing public awareness, improving access through community-based care, and utilizing pharmacists to administer the vaccine could help increase pneumococcal vaccine uptake among older adults [29–31]. Lower vaccination rates among people residing in rural areas were also observed, reflecting a similar trend from our previously published analysis [17]. A study conducted in the US, also reported disparities in pneumococcal vaccination coverage between rural and urban populations, with rural residents being significantly less likely to receive vaccination [32].

Even among those who reported being vaccinated for influenza within the previous year, pneumococcal vaccination coverage remained suboptimal. While 68.4% of individuals aged ≥65 years reported pneumococcal vaccination, only 28.3% of those aged 49–64 years with at least one CMC were vaccinated, highlighting a substantial gap in pneumococcal vaccine coverage among younger high-risk populations. Further, 41.7% of those aged ≥65 and 79.7% of those aged 49–64 with CMCs reported non-vaccination despite recent contact with a family doctor. These findings reflect similar results from other studies indicating persistent missed vaccination opportunities, potentially due to inconsistent vaccine availability, low awareness among both patients and providers, and mixed vaccine acceptance [17,33–35]. Knowledge and awareness remain critical for uptake; prior research has demonstrated that healthcare providers' recommendations significantly enhance vaccination rates [15,34]. Interestingly, higher income within the 49–64 age group with CMCs was associated with lower vaccination, challenging prior assumptions and suggesting that barriers to vaccination in this group extend beyond affordability [25,36]. The potential reasons for this may stem from differences in risk perception, health-seeking behavior, and health beliefs.

Among newly vaccinated participants, those whose CMC status changed between FUP1 and FUP2 had higher vaccination odds. Similar trends were also discussed in other studies, suggesting that increased healthcare interactions related to their underlying condition(s) may have prompted vaccination recommendations [28,37]. Contrary to general trends, higher education was linked to lower vaccination odds among those aged 49–65 with CMCs, diverging from existing research suggesting positive correlations between education and vaccination [25,30]. This unexpected finding warrants further exploration to uncover potential mediating factors in vaccination decision-making within this demographic.

Recent findings highlight the need for public health interventions tailored to under-vaccinated populations – especially older males, rural residents, racialized groups, and residents of specific provinces. Reaching the national target of 80% pneumococcal vaccination coverage will require policy adjustments to enhance vaccine access, e.g., through mobile vaccination clinics in rural areas and strengthened provincial partnerships. The integration of pneumococcal vaccination within established influenza immunization programs could increase coverage among older adults and other high-risk groups [38]. The recent introduction of pneumococcal vaccines with broader serotype coverage offers a promising opportunity to increase uptake rates [39]. Emphasizing the added protection of these vaccines could be a persuasive component in communication strategies by healthcare providers, who play a critical role in raising awareness and addressing access barriers. Tailored communication efforts to inform the public and providers about these more effective vaccines will be crucial in promoting acceptance and understanding of their benefits.

The study has several strengths. Utilizing a comprehensive national dataset allowed for broad geographic and demographic representation, providing valuable insights into vaccination patterns across diverse groups. The robust longitudinal data from the CLSA enabled tracking changes in vaccination coverage over time, offering an opportunity to identify factors influencing vaccination coverage between follow-up waves. However, there are limitations. A primary limitation is that pneumococcal vaccination status was queried only among a subset of FUP2 participants, forcing us to restrict the analysis to the Comprehensive Cohort and potentially limiting the generalizability of the findings. However, it should be noted that our analysis of FUP1 data restricted to the Comprehensive cohort closely aligns with our published findings based on the analysis of the Comprehensive and Tracking cohorts combined. The reliance on self-reported vaccination status may also introduce misclassification due to recall bias, though in our previous analyses using CLSA data, we demonstrated that the degree of accuracy of self-reported vaccination status was unlikely to significantly affect the results even in extreme cases of recall bias [17]. Another limitation is the higher proportion of missing income data in FUP2, which may reduce the ability to interpret how vaccination patterns varied by income level. This limitation was addressed through sensitivity analyses based on imputed FUP1 data. Longitudinal studies provide critical insight into changes in vaccine coverage and will be especially important for tracking vaccination patterns as the landscape of pneumococcal vaccination changes. For example, longitudinal studies can be leveraged following the introduction of new vaccines with broader serotype coverage, such as Pneu-C-21 (CAPVAXIVE™) and Pneu-C-20 (Prevnar®20), to monitor coverage, population health outcomes, and healthcare utilization over time. Additionally, examining vaccination attitudes among

an aging population eligible for pneumococcal vaccination could provide insights into improving communication strategies tailored to older adults. Identifying factors influencing vaccine acceptance in this demographic could inform interventions designed to address specific concerns, overcome hesitancy, and ultimately enhance vaccination rates across Canada. Future research could also explore how provincial vaccination policies, including differences in funding and delivery mechanisms, interact with both system-level and patient-level barriers to influence vaccine uptake, as well as the social, behavioral, and structural factors that may contribute to sex-related differences in vaccination coverage.

## Conclusion

Suboptimal pneumococcal vaccination rates in Canada among adults at risk of IPD present a significant public health challenge with far-reaching implications for health and healthcare systems. Increasing vaccination coverage decreases the risk of IPD, thus reducing the burden that treating IPD cases places on limited healthcare resources. Addressing these gaps requires a thorough understanding of vaccination coverage patterns and trends over time. Continuous surveillance of vaccination rates, particularly among groups with the lowest vaccination coverage, like older males and rural residents, is crucial for evaluating the effectiveness of current interventions and informing future strategies and novel interventions. This ongoing monitoring will be crucial for understanding how vaccination patterns change and assessing the impact of public health efforts to increase vaccine coverage.

## Supporting information

**S1 Table. Description of study variables.** For each variable, we report the corresponding survey question and Canadian Longitudinal Study on Aging (CLSA) variable name, response options offered to participants during the survey, and categorization used for the purpose of this study.
(PDF)

**S2 Table. Self-reported pneumococcal vaccination status (vaccinated or unvaccinated during lifetime) among Canadian Longitudinal Study on Aging (CLSA) cohort participants who were considered eligible to receive a pneumococcal vaccine as per Canada's National Advisory Committee on Immunization (NACI) guidelines, by key sociodemographic characteristics for the period of FUP1 (2015–2018).**
(PDF)

**S3 Table. Distribution of chronic medical conditions (CMC) among individuals eligible for pneumococcal vaccination, by self-reported pneumococcal vaccination status (vaccinated or unvaccinated during lifetime) for the period of FUP1 (2015–2018) Counts, percentages, and 95% confidence intervals within variable strata are shown for two subgroups of interest: 1) individuals aged 65 and older (n = 13,366), and 2) individuals aged 47–64 reported at least one chronic medical condition (CMC) among those listed in the table (cardiovascular disease, chronic lung disease, cerebrovascular disease, chronic kidney disease, diabetes mellitus, cancer, chronic neurologic condition) (n = 6,714).**
(PDF)

**S4 Table. Distribution of chronic medical conditions (CMC) among individuals eligible for pneumococcal vaccination, by self-reported pneumococcal vaccination status (vaccinated or unvaccinated during lifetime) for the period of FUP2 (2018–2021) Counts, percentages, and 95% confidence intervals within variable strata are shown for two subgroups of interest: 1) individuals aged 65 and older (n = 10,530), and 2) individuals aged 49–64 reported at least one chronic medical condition (CMC) among those listed in the table (cardiovascular disease, chronic lung disease, cerebrovascular disease, chronic kidney disease, diabetes mellitus, cancer, chronic neurologic condition) (n = 4,055).**
(PDF)

**S5 Table. Distribution of participants who were asked about pneumococcal vaccination compared to those who were not asked about pneumococcal vaccination during FUP2.**
(PDF)

**S6 Table. Results of sensitivity analyses of factors associated with pneumococcal non-vaccination among 3360 Canadian Longitudinal Study on Aging (CLSA) participants aged 65 years and above to account for change in income status.**
(PDF)

**S7 Table. Results of sensitivity analyses of factors associated with pneumococcal non-vaccination among 2725 Canadian Longitudinal Study on Aging (CLSA) participants aged 47–64 years reported ≥1 CMC to account for the change in income status.**
(PDF)

**S8 Table. STROBE checklist for cross-sectional studies: completed checklist outlining adherence to reporting guidelines for observational studies using the STROBE (Strengthening the Reporting of Observational Studies in Epidemiology) statement.**
(PDF)

**S1 Fig. Canadian Longitudinal Study on Aging (CLSA) participants' flowchart for FUP2 and inclusion into our analyses as relevant.**
(TIF)

**S2 Fig. Change in income status of CLSA participants eligible for pneumococcal vaccination from the period of FUP1 (2015–2018) to FUP2 (2018–2021).**
(TIF)

## Acknowledgments

This research was made possible using the data collected by the Canadian Longitudinal Study on Aging (CLSA), led by Drs. Parminder Raina, Christina Wolfson, and Susan Kirkland. This research has been conducted using the Comprehensive Baseline v7.0, Comprehensive Follow-up 1 v4.0, and Comprehensive Follow-up 2 v1.0 data under Application Number 2206026. The opinions expressed in this manuscript are the author's own and do not reflect the views of the Canadian Longitudinal Study on Aging. The authors gratefully acknowledge the time and commitment of the CLSA participants, without whom this research would not be possible.

## Author contributions

**Conceptualization:** Giorgia Sulis, Christina Wolfson, Nicole E. Basta.

**Data curation:** Giorgia Sulis, Nawal Maredia.

**Formal analysis:** Nawal Maredia.

**Funding acquisition:** Giorgia Sulis, Nicole E. Basta.

**Methodology:** Giorgia Sulis, Nicole E. Basta.

**Project administration:** Nicole E. Basta.

**Supervision:** Giorgia Sulis, Nicole E. Basta.

**Validation:** Giorgia Sulis, Nawal Maredia.

**Visualization:** Nawal Maredia.

**Writing – original draft:** Giorgia Sulis, Nawal Maredia.

**Writing – review & editing:** Giorgia Sulis, Nawal Maredia, Christina Wolfson, Nicole E. Basta.

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
