## [Decision Letter · Decision Letter 0]

6 Jul 2025

Dear Dr. Sulis,

Thank you for submitting your manuscript to PLOS ONE. After careful consideration, we feel that it has merit but does not fully meet PLOS ONE’s publication criteria as it currently stands. Therefore, we invite you to submit a revised version of the manuscript that addresses the points raised during the review process.

We look forward to receiving your revised manuscript.

Kind regards,

Dominic Luke Thorrington, PhD

Academic Editor

PLOS ONE

2. For studies involving third-party data, we encourage authors to share any data specific to their analyses that they can legally distribute. PLOS recognizes, however, that authors may be using third-party data they do not have the rights to share. When third-party data cannot be publicly shared, authors must provide all information necessary for interested researchers to apply to gain access to the data. (https://journals.plos.org/plosone/s/data-availability#loc-acceptable-data-access-restrictions)

3. Please include a caption for figure 1 and 2.

Additional Editor Comments:

We have received sufficient peer review reports on your manuscriptand we would like to share a summary of the feedback with you. Overall, the reviewers found your work timely and policy-relevant.

The reviewers have raised several important points that require your attention, particularly concerning the study design, analysis, and interpretation of findings. The most crucial points are highlighted below:

Major Points to address based on the reviews:

Study Design and Data Clarification

Selection Bias and Missing Data

Statistical Analysis and Survey Weights

Model Validation and Goodness-of-Fit

Impact of COVID-19 Pandemic

Incorporation of Physician Contact

There are several other smaller comments but the major points are highlighted above.

We encourage you to carefully consider all feedback provided by the reviewers. Please revise your manuscript accordingly and provide a point-by-point response to each comment in your resubmission.

We look forward to receiving your revised manuscript.

Reviewers' comments:

Reviewer's Responses to Questions

**Comments to the Author**

1. Is the manuscript technically sound, and do the data support the conclusions?

Reviewer #1: Yes

Reviewer #2: Yes

Reviewer #3: Yes

Reviewer #4: Partly

2. Has the statistical analysis been performed appropriately and rigorously?

Reviewer #1: Yes

Reviewer #2: No

Reviewer #3: I Don't Know

Reviewer #4: Yes

3. Have the authors made all data underlying the findings in their manuscript fully available?

Reviewer #1: No

Reviewer #2: No

Reviewer #3: Yes

Reviewer #4: Yes

4. Is the manuscript presented in an intelligible fashion and written in standard English?

Reviewer #1: Yes

Reviewer #2: Yes

Reviewer #3: Yes

Reviewer #4: Yes

Reviewer #1: The discussion briefly touches on provincial differences, but the paper could be strengthened by explaining how the findings might inform region-specific strategies, especially in provinces like Newfoundland where vaccination rates were particularly low.

Since some of the FUP2 data collection took place during the COVID-19 pandemic, it would be useful to add a short reflection on how this period may have affected vaccination behavior or access to care. This context could help clarify some of the observed trends.

It may be worth including a brief note to confirm that multicollinearity among predictors was checked and not found to be an issue. This would improve confidence in the stability of the regression estimates.

While casewise deletion is reasonable given the low rate of missing data, a short explanation of why imputation was not used would demonstrate careful consideration of missingness, particularly in the case of income.

Reviewer #2: This manuscript used the newly released FUP2 CLSA data to study the uptake of pneumococcal vaccination and to identify factors associated with vaccination uptake compared with data from FUP1. The results are timely and policy relevant. Overall, this is a nicely written paper. The review wishes to see a bit more clarification on the study design and analysis, in particular with respect to selection bias, complete-case analysis, assessment of effect modification, and the choice of not using CLSA survey weights yet attempting to provide inference and conclusion aimed at national level estimates. I have enclosed detailed feedback below.

Major comments

1. On longitudinal data. Can you clarify “non-overlapping follow-up surveys” post baseline? Does this mean that different individuals were surveyed at each time point? It’s a bit confusing here if non-overlapping is referring to non-overlapping participants or non-overlapping periods. The reviewer wishes to see clarification on whether the survey is done longitudinally, which features data collection on survey participants repeatedly. The reviewer does not think the data can be categorized as repeated cross-sectional data; it should just be termed as longitudinal data.

a. The longitudinal data needs clarification, although age is not collected at FUP2 (exclude 485 participants) was age collected at baseline and FUP1?

b. Again, clarification is needed as more than half of the FUP2 participants were excluded from the study cohort.

2. On selection bias.

a. Participants with missing data on self-reported pneumococcal vaccination status were excluded. Further inducing selection bias. Please justify the decision to exclude missing responses. Missing responses can be associated with vaccine uptake (that is informative missing), which may or may not be flagged by simply looking at demographic differences between the patient subgroups (as done in the sensitivity analysis). The reviewer suggests running a weighted logistic regression, weighted by the probability of missing self-reported vaccination status. This should be included as a sensitivity analysis.

3. Choice of statistical analysis for survey data

a. Please justify the rationale for not fitting the regression model properly, adjusting for the survey weights provided in the CLSA datasets to obtain national level estimates.

b. Please clarify what “logit transformation of proportions” means here. Are these univariate analyses by fitting multiple univariate logistic regressions for each of the factors by vaccination exposure? Again, see comments above, why not using survey weights here. If this was indeed the univariate analysis, please write the accurate statistical analysis term here.

c. Please justify the decision of not testing any interaction effect where the association between key sociodemographic factors can be moderated by history of chronic conditions as well as vaccination pattern and health care utilization in the previous 12 months.

d. Despite the large sample size (>10,000 total), fig1 and fig2 demonstrate wide confidence intervals for many covariates. Are there any model goodness-of-fit checks? Please provide a light analysis demonstrating the validity of the fitted logistic regression model.

Minor comments

1. Consider changing the manuscript title to use of follow-up 2 to second wave. Follow-up 2 is a CLSA specific term where survey wave is generally accepted in wider context.

2. In the introduction section, it would be helpful to introduce evidence on vaccine uptake globally.

Reviewer #3: My main Comments (Minor Revisions Recommended):

- On page 2, the manuscript asks, “Did you receive funding for this work?” but this question is left unanswered. However, funding is mentioned later on page 28. Please ensure that the funding information is clearly and consistently reported.

- In the Introduction, when mentioning the second aim to identify factors associated with being newly vaccinated, I recommend specifying that these factors refer to sociodemographic variables to better align with the analysis, findings, and discussion.

- Regarding the use of CLSA data, please clarify if the researchers applied for access and agreed to abide by the CLSA’s Data and Sample Access Policy.

- Please specify which software was used for data analysis.

- The manuscript states:

“The proportion of CLSA participants who were asked questions about pneumococcal vaccination was lower at FUP2 compared to FUP1. This decrease was due to a temporary, unexplained omission of certain sections of the survey questionnaire between early 2018 and early 2019.” Please clarify what is meant by “unexplained omission” in this context.

- The finding that males consistently report lower vaccination rates compared to females requires further explanation. Could this be due to social norms, personal beliefs, or other factors? A brief discussion is recommended.

- Interestingly, higher income within the 49–64 age group with CMCs was found to be associated with lower vaccination rates, challenging prior assumptions that affordability is a primary barrier. Please discuss possible reasons why higher-income individuals in this group might have lower vaccination uptake.

- Contrary to general trends, higher education was found to be linked to lower vaccination odds among those aged 49–65 with CMCs, diverging from existing research suggesting positive correlations between education and vaccination. Please provide a clearer explanation or discussion of why this unexpected finding might occur.

Reviewer #4: Comments:

1. Abstract section is should be improved by focusing the significant statistical results of the study. The introduction section can be improved to focus on specific area of the study and authors may add novelty statement.

2. Author are suggested to improve the methodology section. Since the FUP2 vaccine questions were only asked of participants in the Comprehensive cohort, the restriction may lead to selection bias. Author should elaborate on how representative this subset is compared to the full CLSA cohort and whether weighting was used to address this.

3. The total eligible sample changes across aims (e.g., 10,530 vs. 3,733 for ≥65). Authors are suggested to include a flow diagram /CONSORT-style figure to clearly depict inclusion/exclusion criteria. Authors are suggested to include the STROBE check list (as an Appendix), and verify if they are complying with all the items on the STROBE check list.

4. The regression models include many sociodemographic covariates. Authors are suggested to clarify whether multicollinearity diagnostics (e.g., VIF) were performed and if model assumptions (e.g., goodness-of-fit) were validated.

5. Data collection for FUP2 occurred during the COVID-19 pandemic (Lines 412–415). This may have artificially lowered vaccination rates due to healthcare access disruptions. A stratified analysis (e.g., pre- vs. post-2020) would strengthen the findings and help isolate pandemic-related effects.

6. Although physician contact in the previous 12 months is noted (e.g., Lines 209–210, Table 1), its role is not explicitly modeled as a predictor of new vaccination. Given the public health relevance, authors should consider incorporating this variable into the adjusted models.

7. Authors are suggested to improve the discussion by linking findings to provincial vaccination policies (e.g., funding, public health messaging) and differentiating between system-level vs. patient-level barriers.

8. Authors should describe any limitations of their study. Furthermore, these limitations should be discussed.

9. The manuscript is generally well-written, however, authors are suggested to improve the English language. There are minor grammatical inconsistencies and typographical errors especially in the Discussion and Conclusion section.

10. Provide abbreviations used.

**Do you want your identity to be public for this peer review?** For information about this choice, including consent withdrawal, please see our Privacy Policy

Reviewer #1: No

Reviewer #2: No

Reviewer #3: No

Reviewer #4: **Yes:** Dr. Mehmood Ahmad

---

## [Author Response · Author response to Decision Letter 1]

29 Jul 2025

Please see attached file "CLSA FUP2 Pneumo vax_Sulis et al_Response to Reviewers".

---

## [Decision Letter · Decision Letter 1]

19 Nov 2025

Changes in pneumococcal vaccine coverage in the Canadian Longitudinal Study on Aging (CLSA): An analysis based on the 2018-2021 follow-up 2 survey

PONE-D-25-27258R1

Dear Dr. Sulis,

We’re pleased to inform you that your manuscript has been judged scientifically suitable for publication and will be formally accepted for publication once it meets all outstanding technical requirements.

Kind regards,

David J. Diemert, M.D.

Academic Editor

PLOS ONE

Additional Editor Comments (optional):

Reviewers' comments:

Reviewer's Responses to Questions

**Comments to the Author**

Reviewer #1: All comments have been addressed

Reviewer #4: All comments have been addressed

2. Is the manuscript technically sound, and do the data support the conclusions?

Reviewer #1: Yes

Reviewer #4: Partly

3. Has the statistical analysis been performed appropriately and rigorously?

Reviewer #1: Yes

Reviewer #4: Yes

4. Have the authors made all data underlying the findings in their manuscript fully available?

Reviewer #1: Yes

Reviewer #4: Yes

5. Is the manuscript presented in an intelligible fashion and written in standard English?

Reviewer #1: Yes

Reviewer #4: Yes

Reviewer #1: (No Response)

Reviewer #4: Authors have diligently addressed almost all the comments and concerns raised during the review process. The revisions made have significantly improved the quality and clarity of the article.

**Do you want your identity to be public for this peer review?** For information about this choice, including consent withdrawal, please see our Privacy Policy

Reviewer #1: No

Reviewer #4: **Yes:** Mehmood Ahmad

---

## [Editor Report · Acceptance letter]

PONE-D-25-27258R1

PLOS One

Dear Dr. Sulis,

I'm pleased to inform you that your manuscript has been deemed suitable for publication in PLOS One. Congratulations! Your manuscript is now being handed over to our production team.

Kind regards,

on behalf of

Dr. David J. Diemert

Academic Editor

PLOS One